# Self-poled piezoelectric polymer composites *via* melt-state energy implantation

Zhao-Xia Huang [1,2] ✉, Lan-Wei Li[1,2], Yun-Zhi Huang[1], Wen-Xu Rao[1], Hao-Wei Jiang[1], Jin Wang [1], Huan-Huan Zhang[1], He-Zhi He[1] & Jin-Ping Qu [1] ✉

Lightweight flexible piezoelectric polymers are demanded for various applications. However, the low instinctively piezoelectric coefficient (*i.e.* d33) and complex poling process greatly resist their applications. Herein, we show that introducing dynamic pressure during fabrication is capable for poling polyvinylidene difluoride/barium titanate (PVDF/BTO) composites with d33 of ~51.20 pC/N at low density of ~0.64 g/cm$^3$. The melt-state dynamic pressure driven energy implantation induces structure evolutions of both PVDF and BTO are demonstrated as reasons for self-poling. Then, the porous material is employed as pressure sensor with a high output of ~20.0 V and sensitivity of ~132.87 mV/kPa. Besides, the energy harvesting experiment suggests power density of ~58.7 mW/m$^2$ can be achieved for 10 N pressure with a long-term durability. In summary, we not only provide a high performance lightweight, flexible piezoelectric polymer composite towards sustainable self-powered sensing and energy harvesting, but also pave an avenue for electrical-free fabrication of piezoelectric polymers.

Piezoelectric materials can convert mechanical deformation into electricity, which makes them desirable candidates for self-powered sensors, energy harvesters, etc[1,2]. Moreover, by using the reverse piezoelectric effect, it could transfer the input electrical signal into mechanical deformation and serve as actuators[2]. Comparing with traditional piezoelectric ceramics like barium titanate (BTO) and lead zirconate titanate (PZT), polymers show considerable advantages of being lightweight, flexible, and easy-to-process[3,4]. However, the main drawback of piezoelectric polymers is their low piezoelectric coefficient (d33, in this work, used as an absolute value). For instance, the d33 for well electrical-poled polyvinylidene difluoride (PVDF) is around ~23 pC/N and is ~40 pC/N for the expensive PVDF-copolymers (i.e., PVDF-TrFE)[2,5]. Although the lower density of PVDF (1.78 g/cm$^3$) than conventionally used BTO (6.02 g/cm$^3$) is beneficial for self-powered sensors, such a low d33 value is still resisting the application of piezoelectric polymers[2,3,6,7].

To enhance the d33 of polymers, it is found that including piezoelectric ceramics and electrical poling could be a vital approach[6,8–10]. However, such achievement was based on the high loading of ceramics like BTO (usually higher than 30 wt%), which therefore impairs the lightweight and flexibility of polymers[11,12]. To this end, previously, we showed that through selectively localizing BTO on the pore surface of open-cellular structured PVDF-based ferroelectrets could largely enhance its piezoelectric output, with a low density[13]. Moreover, the space charge electret effect was demonstrated as the major reason for the piezoelectric performance enhancement. However, even though lightweight flexible ferroelectret polymers can be obtained using such a concept, it still relies on the energy-intensive and time-consuming electrical-poling process for the formations of ferroelectrics, which may cost more energy than it can harvest[14,15]. Thus, from the sustainable development viewpoint, fabricating piezoelectric materials without the requirement of electrical-poling attracted attention.

[1]National Engineering Research Center of Novel Equipment for Polymer Processing, Key Laboratory of Polymer Processing Engineering, Ministry of Education, Guangdong Provincial Key Laboratory of Technique and Equipment for Macromolecular Advanced Manufacturing, Department of Mechanical and Automotive Engineering, South China University of Technology, 510641 Guangzhou, China. [2]These authors contributed equally: Zhao-Xia Huang, Lan-Wei Li. ✉e-mail: mehuangzx@scut.edu.cn; jpqu@scut.edu.cn

To avoid electrical poling, fillers like MXene and carbon nanotubes were loaded into PVDF for self-poling, due to its strong intrachain interactions with PVDF molecules that form the spontaneous polarization, after ink printing into devices[16–18]. Although self-poled piezoelectricity could be achieved by blending PVDF with MXene, considering the non-piezoelectric nature of MXene, there will be potential for further enhancement of d33 by introducing piezoelectric fillers. Thus, to further enhance the piezoelectric properties of lightweight flexible polymers, it is essential to gain deeper insight into their dipolar alignment behavior, and the inter- and intra-chain interactions between the polymer and the included piezoelectric fillers.

In this work, through employing the widely employed and well-investigated porous PVDF/BTO foam (PBf) as model material, we proposed a melt-state dynamic pressure procedure that is capable of fabricating the self-poled piezoelectric. The as-prepared pristine PBf shows a high d33 of ~51.20 pC/N, at a relatively low density of 0.64 g/cm³. The mechanism of melt-state energy implantation-induced self-poling of both PVDF and BTO components is investigated. Moreover, we showed that the PVDF/BTO foam can also serve as a highly sensitive pressure sensor and high-output mechanical energy harvester.

## Results

### PBf fabrication under melt-state energy implantation

As shown in Fig. 1a, the PBfs were fabricated via a three-step process including melt-blending, compression molding, and salt leaching. In the first step, weighted PVDF, BTO, and NaCl (21:9:70 wt%) were melt blended using a Brabender mixer at 195 °C to achieve a fine distribution and dispersion of BTO and NaCl in the PVDF matrix. Consequently, the mixed compounds were compression molded into specimens with 1 mm thickness at the same temperature (195 °C). In this step, the conventional art was using a constant pressure (25 MPa, selected based on pre-experiment) for 6 min, and the resultant sample was named as CP-PBf. However, for our method, cycled dynamic pressure (120 repeats) was applied (the obtained sample was named as EI-PBf), and the pressure-time relationship of each unit is shown in Fig. 1c. Meanwhile, the pressure-time relationship for compression molding CP-PBf was shown in Supplementary Fig. 1. After compression molded, all spacemen were immersed in deionized water at 60 °C for selective removal of NaCl to obtain the open-cellular structure. Figure 1b shows the photographs of PBf before and after salt leaching, and the salt-free one reveals good flexibility. In addition, we also measured the

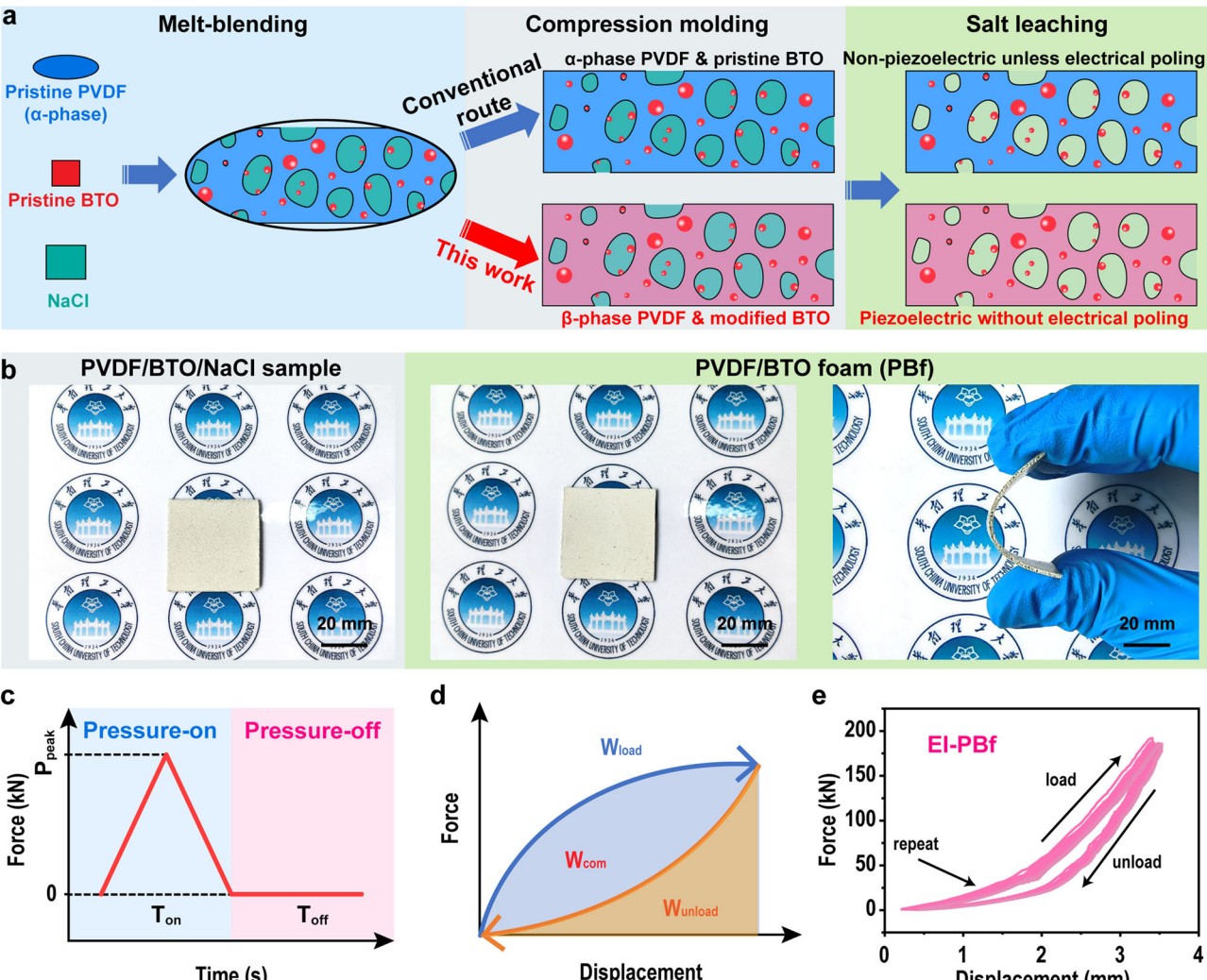

**Fig. 1 | Energy implantation technology and PBfs manufacturing. a** Schematic of the fabrication procedure of PBf samples. **b** Photographs of PVDF/BTO/NaCl samples before and after salt leaching, and the salt-free sample shows good flexibility. **c** Illustration of the force-time relationship of one cycle in our melt-state dynamic pressure process. **d** Illustration of the forced displacement in each cycle in our method, areas under the loading and unloading curves are considered as the work of loading ($W_{load}$) and unloading ($W_{unload}$), where the difference between them is the work of compression ($W_{com}$). **e** The force-displacement curves during the entire compression molding process (from the 1st cycle to the 120th cycle) of EI-PBf.

densities, porosities, and pore sizes of all samples fabricated as shown in Supplementary Figs. 2, 3, and 4.

From the force-time curve shown in Fig. 1c, it is clear that the EI method consists of two pressure regions: pressure-on and pressure-off. In the pressure-on region, the pressure was applied on the material which could thin the sample, and energy was implanted. While in the pressure-off region, the thickness of the sample will be increased by relaxing it without the application of pressure, which leaves room for the next round of the pressure-on process to thin it again (see Supplementary Fig. 5). Herein, $T_{off}$, the period of the pressure-off region, was set as 1.20 s.

Then, as shown in the schematic in Fig. 1d, in each unit, a loading process and an unloading process are included, while the energy ($W_{load}$) was applied to the melt during the loading process, and part of the energy ($W_{unload}$) was relaxed during the unloading process. Consequently, the difference between $W_{load}$ and $W_{unload}$ is the implanted energy ($W_{com}$) injected that is stored in the melt, and such $W_{com}$ can be accumulated during the cycle treatment. To reveal the energy implantation results during processing, the relationship between applied pressure and the variations in EI sample thickness was monitored as shown in Fig. 1e, where the area under a compression force-delta thickness is considered as energy. Moreover, we also recorded the force-delta thickness of CP-PBf in Supplementary Fig. 6. Detailed discussion of the energy implantation behavior and its effect on the samples will be given later.

## Structural evolution of PBf induced by energy implantation

Figure 2a shows the scanning electron microscopy (SEM) images of as-prepared EI-PBfs. The morphology of the CP sample was also revealed in Supplementary Fig. 7. In these SEM figures, we can see that both samples present similar porous structures with pore sizes from ~20 μm to ~500 μm. Moreover, it is clear that BTO particles are mainly located on the pores' surfaces, as identified by energy dispersive spectroscopy (EDS) mappings (Fig. 2a). Moreover, the results also suggest that the EI method does not change the open-cellular structure of PBfs. Then, Fourier-transform infrared spectroscopy (FTIR) was used to investigate the influence of implanted energy on the crystalline structure of PVDF, as shown in Fig. 2b. In these FTIR spectra, peaks around 766 cm$^{-1}$ and 840 cm$^{-1}$ are assigned to the non-electroactive phase (α-crystal) and electroactive phase (β and γ-crystals) of PVDF, respectively[6]. Moreover, as been widely investigated, compared with the γ-crystal, the β-phase PVDF shows better piezoelectricity. Then, peaks in 1234 cm$^{-1}$ and 1275 cm$^{-1}$ are used to distinguish the β and γ-crystals[6,19] (more information can be found in Supplementary Information). Consequently, we calculated the relative fraction of β-crystals of both PBfs via energy implantation and conventional method, as shown in Fig. 2c. From the results, we can see that the implanted energy can obviously induce the formation of β-crystal in PVDF. The reason for the energy-induced crystallization can be considered due to the more energy injected that would press the PVDF chains into a denser packing state, where β-crystal shows the highest packing density among all polymorphs of PVDF[19]. Moreover, compared with the widely employed stretch-induced β-phase formation[20], our method based on compression can be more efficient. In addition to FTIR, X-ray diffraction (XRD) was also employed to reveal the crystalline structure of each sample. As shown in Fig. 2d, the intensities of peaks belonging to BTO are much higher than that of PVDF, due to the higher level of crystallinity of BTO. In Fig. 2e, f, we showed the enlarged XRD patterns around the PVDF range, and their peaks fitting. From the XRD data, it is clear that an additional peak around ~20.3° shows in EI-PBf, which is assigned to the (110)/(200) plane of β-PVDF, in line with the FTIR results[6]. More discussions on the dynamic pressure-induced crystalline structure variations can be found in our previous report[21]. In addition to the crystalline structure, the crystallinity of a polymer also has a significant influence on its performance. Thus, the crystallinities of samples

fabricated via different methods were also measured as shown in Supplementary Fig. 8.

Besides the crystalline structure of PVDF, we also investigated the structure evolution of BTO, as shown in Fig. 2g. As suggested, peaks around ~31.4°, 31.7°, and 38.8°, can be assigned to the (101), (110), and (111) planes of BTO[22]. From the data, we can see that after compression, there is a crystal plane rotation present in the EI sample. As identified in the enlarged XRD in Fig. 2g, h, we can see the intensities of (101) and (110) planes for CP- and EI-PBf samples are different. For the EI sample, the (110) plane shows stronger intensity than the (101) plane, while the CP-sample shows reverse phenomenon. It suggests that the EI sample has more BTO with (110) plane aligned in the in-plane direction. Moreover, from the enlarged data, it is clear there are several levels of the shift in the 2-theta of the peaks.

## Piezoelectric properties of PBfs

Next, we measured the electrical responses of as-prepared PBfs under pressure. Figure 3a shows the general mechanism and process of the piezoelectric effect, in which the pressure applied on the material can induce the dipole movement and generate an electrical potential difference between the up and down surfaces. In Fig. 3b, we showed the transferred charge between two electrodes on each surface of PBfs, as a function of pressure applied (the photographs of the setup we used for measurements were shown in Supplementary Fig. 9a). All samples shown in Fig. 3b were repeated three times, with the data collected in Supplementary Fig. 10. Herein, to avoid the triboelectric contribution during measurement, a directly extracting method recently proposed[23] was employed (see Supplementary Information). Besides, a commercial-available well-poled piezoelectric PVDF film (from PolyK Technologies) and a non-piezoelectric polyethylene terephthalate (PET) film were utilized to validate this method (see Supplementary Fig. 11). Consequently, the slope of charge vs force is calculated as d33. In addition, we also collected the force and displacement vs time data in Supplementary Fig. 9b for information. From the data, the d33 values for CM-PBf and EI-PBf samples are 0.58 pC/N and 51.20 pC/N, respectively. The d33 of 0.58 pC/N for the CM-PBf sample is reasonable and widely reported since no electrical poling was applied[2]. While, for the EI-PBf sample, a high d33 is clearly evidenced, demonstrating that the energy implantation could be a poling technique. Moreover, it should be mentioned that even though electrical poling is introduced, its d33 (-16.3 pC/N, see Supplementary Fig. 12) is still lower than the d33 of our EI-PBf. Besides, another widely employed method for d33 measurement as well as a d33 meter was also used to validate our results, as shown in Supplementary Figs. 13 and 14, which showed similar d33 values of our EI-PBf.

In addition, we also recorded the open-circuit voltage ($V_{oc}$) and short-circuit current ($I_{sc}$) under different pressures in Fig. 3c, d, respectively, for the purpose of self-powered pressure sensing. From the $V_{oc}$-pressure curves, we can see that our EI sample could generate a voltage from ~10 V to 20 V, depending on the pressure applied from 14.4 kPa to 86.6 kPa. The pressure sensitivity is calculated as 132.87 mV/kPa. A similar trend is also observed in the $I_{sc}$-pressure, where the $I_{sc}$ increased from 2.78 μA to 3.28 μA, depending on the pressure from 14.4 kPa to 86.6 kPa. Such a high sensitivity evidenced that our EI sample could be a promising self-powered pressure sensor. Furthermore, we also extended the applied pressure up to 158.7 kPa, in which the results still showed a good linear relationship along with the high pressure applied (see Supplementary Fig. 15).

In addition to the quantity of d33, several other values, including g33 and figure-of-merit (FOM), are also essential for evaluating a piezoelectric material[2]. The g33 value can be calculated as $g_{33} = d_{33}/\varepsilon_0 k_{33}$, where $\varepsilon_0$ is the permittivity of vacuum, and $k_{33}$ is the relative permittivity in the 33 direction. Herein, the dielectric properties of our sample were monitored for obtaining k33, as shown in Supplementary Fig. 16. Moreover, considering the demand of obtaining lightweight

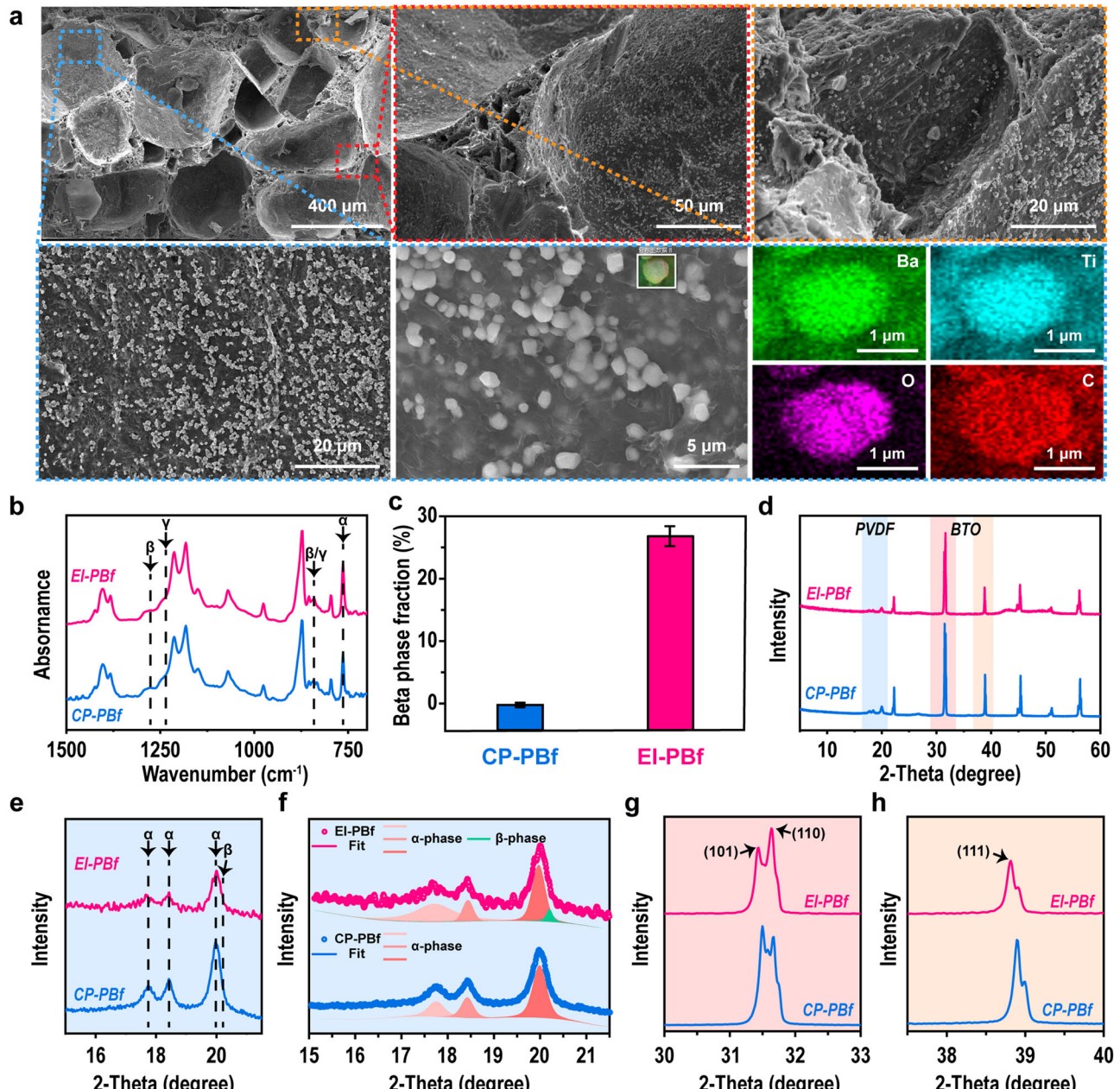

**Fig. 2 | Structural characterizations of PBfs. a** SEM images of as-prepared EI-PBf at different magnifications, and the corresponding EDS mapping. **b** FTIR spectra of CP- and EI-PBf with wavenumber ranges from 700 cm⁻¹ to 1500 cm⁻¹. **c** Calculated β-phase PVDF fraction of PBf fabricated via both CP and EI. **d** XRD diffractions of CP- and EI-PBf. **e, f** Enlarged XRD from 15° to 22°, and the corresponding peak fitting; **g, h** Enlarged XRD of (101), (110) and (111) planes of BTO. Error bars represent the standard deviations from the statistic results of three repeats.

piezoelectric material for potential applications, the density-specific $g_{33}^{ds}$ as $g_{33}^{ds} = g_{33}/\rho$ was also defined, where $\rho$ is the density of the sample as shown in Supplementary Fig. 2.

Consequently, to reveal the superiority of our method for obtaining high-performance piezoelectric material, we performed a comparison of the $g_{33}^{ds}$ between our sample and several recently reported advanced piezoelectric materials[16,17,24–26], as shown in Fig. 3e (detailed information on these materials is shown in Supplementary Table 1). It should be mentioned that all these reported materials are measured after a time-consuming electrical-poling process, while ours are directly after melt-state fabrication. Moreover, considering the porous feature of our material, the competitors contain not only solid-one but also ceramic-polymer hybrid porous ferroelectrics as shown in Supplementary Fig. 17 and Supplementary Table 2. Although these

porous piezoelectrics can perform better g33 values due to the formation of ferroelectret effects during electrical poling, our sample does not require any post-treatment to present a high g33 value. From the data, it is clear that the $g_{33}^{ds}$ of our mechanical poled sample is 25020 mV/Pa•g, which is much higher than that of these electrical-poled materials, including both polymers and ceramics. Moreover, we also prepared a comparison of both FOM and $g_{33}$[16,17,24,25,27–39] in Fig. 3f. Herein, the FOM is defined as the product of $g_{33}$ and $d_{33}$, and can be considered as a universal quantity for evaluating a piezoelectric material. In Fig. 3f, we can see that even compared with the advanced inorganic- and hybrid ferroelectric-based piezoelectric materials, our method still presents the highest FOM and $g_{33}$, suggesting a high-performance piezoelectric. Besides the piezoelectric properties, the tensile properties, which are also crucial for the application, are also

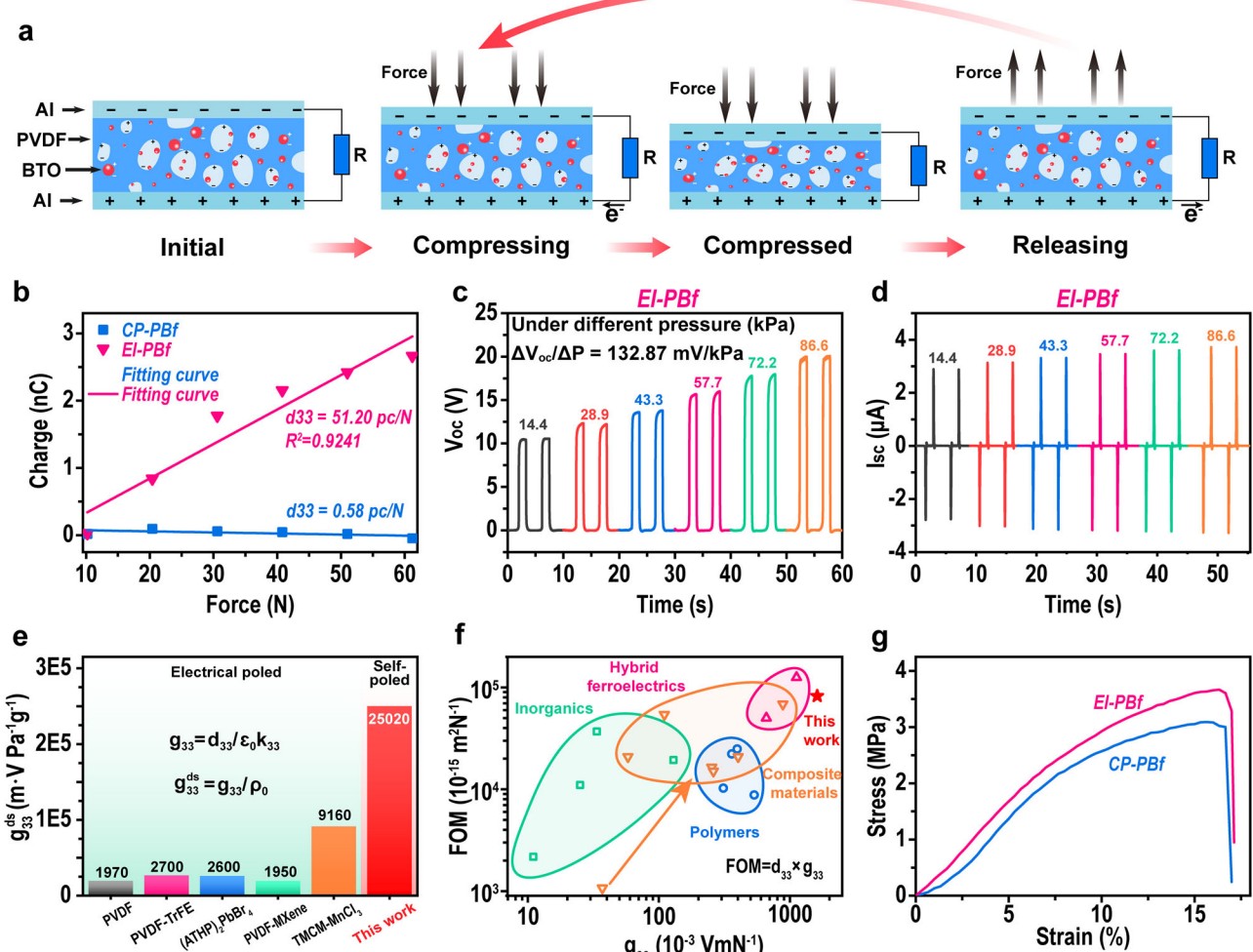

**Fig. 3 | Piezoelectric properties of self-poled PBfs. a** Schematic of the general process and mechanism of piezoelectric effect. **b** Transfer charge as a function of various applied forces applied on the PBfs. **c, d** $V_{oc}$ and $I_{sc}$ of EI-PBf under various pressure applied. **e, f** Comparison between our EI-PBf and other reported piezo-electric in $g_{33}^{ds}$[16,17,24–26], FOM and $g_{33}$[16,17,24,25,27–39]. **g** Tensile strain-stress curves of CP- and EI-PBf.

measured as shown in Fig. 3g, in which the EI-PBf shows better mechanical properties including both strength and stretchability.

**Mechanism of energy implantation-induced poling**

As revealed in Fig. 1a, it is clear that the major feature of the EI method is introducing a pressure-off process in each unit, which can strengthen the energy-accumulating process. Herein, to confirm the contribution of implanted energy to d33, we further fabricated EI-PBfs with various $T_{off}$ (the pressure-time conditions are shown in Fig. 4a) and measured their d33 values. Along with longer $T_{off}$ used during compression molding, as shown in Fig. 4b, c, the d33 value increases from 7.55 pC/N for 0 s $T_{off}$ to 51.20 pC/N for 1.20 s, which is the optimal $T_{off}$ in this work, and then drops to 29.92 pC/N for 2.10 s.

To reveal the reason for the optimal $T_{off}$ presented in d33, we showed the $W_{com}$ of each $T_{off}$ during fabrication as a function of repeats, in Fig. 4d. The detailed force-displacement relationships of all samples are shown in Supplementary Fig. 18. From data, we can see that along the whole fabrication cycles, sample with $T_{off}$ of 1.20 s shows the highest $W_{com}$ than others, suggesting such a $T_{off}$ is the optimal value for the energy accumulation. Moreover, to clearly show the relationship between implanted energy and $T_{off}$, we calculated the summation of the implanted energy of different samples, as shown in Fig. 4e. From data, along with the $T_{off}$ increases from 0 s to 2.10 s, the implanted energy increases from 196.3 ± 13.7 J/g (0 s) to 265.6 ± 19.5 J/g (1.20 s), and then drops to 260.4 ± 18.5 J/g

(2.10 s). Importantly, we also found that the trend in d33-$T_{off}$ shows a clear correlation with the implanted energy-$T_{off}$ (detailed data and fitting can be seen in Supplementary Fig. 19), which clearly confirms the dominating role of energy implantation in the self-poling piezoelectric.

Then, we focused on the mechanism of energy implantation-induced poling of PBfs. For the conventional method obtained foam (CP-PBf in this work), there is no obvious piezoelectric output due to no electrical-poling process used. While, for the EI-PBf, the EI process can induce the structure evolution of both PVDF and BTO, which contributes to the piezoelectric. To quantitatively reveal the contribution of PVDF and BTO, we fabricate the BTO-free samples (Pfs), as shown in Fig. 4f. From the data, we can see that without the contribution of BTO, the EI-Pf shows a lower d33 of ~10.20 pC/N than the one with BTO (51.20 pC/N). It should be mentioned that the d33 of CM-Pf is still low (~1 pC/N), which is in agreement with previous investigations[13]. As been investigated, the alignment chain structure of PVDF, which is the β-crystal, can present piezoelectric properties. The β-phase of each sample with various $T_{off}$ was shown in Supplementary Fig. 20, which shows that the conventional method cannot induce the formation of β-phase PVDF, while the EI could induce the formation of ~20% β-phase. Thus, the reason for obtaining piezoelectric in the PVDF matrix of the EI method prepared is proposed.

Furthermore, through a simple calculation, we can hypothesize the difference in d33 of EI-Pf (10.20 pC/N) and EI-PBf (51.20 pC/N) is

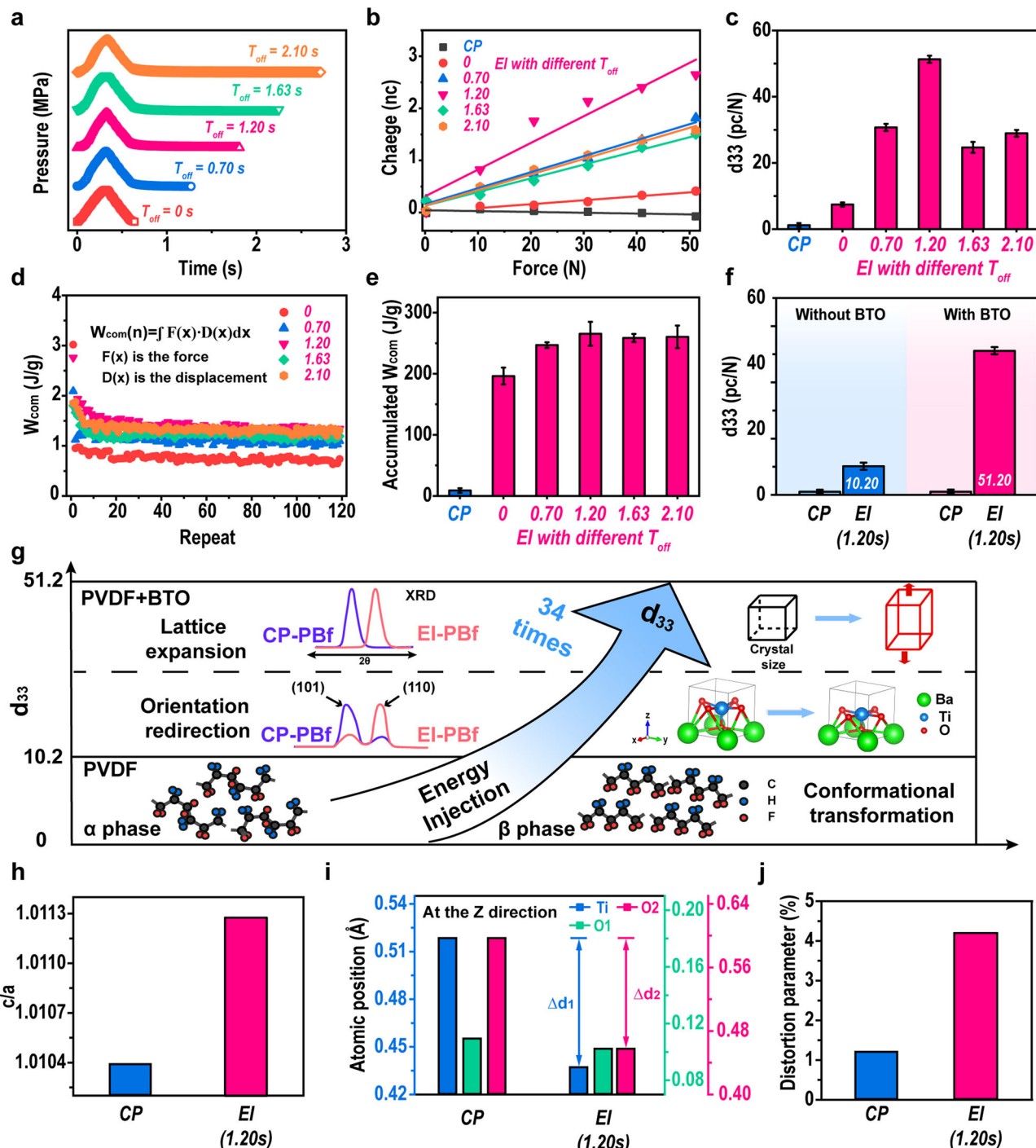

**Fig. 4 | Mechanism of self-poled piezoelectric properties of PBfs induced by energy implantation. a** The pressure-time relationship of repeat unit in EI method with different $T_{off}$. **b** Transferred charge-force relationships of samples fabricated via various approaches. **c** The calculated d33 values of each sample. **d** $W_{com}$ as functions of a repeat of EI samples with various $T_{off}$. **e** Accumulated $W_{com}$ of each sample. **f** d33 values of samples with and without BTO that were fabricated via CP-method and EI method ($T_{off}$ of 1.20 s). **g** Schematic of the self-poling mechanism of EI-PBf. **h, i, j** c/a values, atomic Z-axial position, and distortion parameter of sample fabricated via CP-method and EI method ($T_{off}$ of 1.20 s). Error bars represent the standard deviations from the statistic results of three repeats.

due to BTO. In the XRD measurement of PBf, we showed that the EI method can induce variations in the crystalline structure of BTO. Herein, a Rietveld refinement analysis was performed to obtain detailed information on the BTO structure. Firstly, from the XRD results, it is clear that the energy implantation can promote the orientation of BTO crystals, as higher intensity of the (110) plane than the (101) plane in the EI sample, while for the CP-sample the intensity of (101) plane is much higher than (110) plane. Such results suggest that

more (110) planes in EI-sample redirection to the in-plane direction, which has been investigated to present higher piezoelectric repones[40].

Then, we calculated the lattice parameter (a, b, and c) in Supplementary Table 3, while the c/a value was employed, as shown in Fig. 4h. Compared with the CP-PBf, we can see that EI sample shows the highest c/a value, which has been proved for the piezoelectric properties of BTO[41]. Besides, through Rietveld refinement analysis, the Z-axial positions of atomics were calculated as shown in Fig. 4i. In which,

the data shows a clear trend of the move down of Ti and O atomics between CP- and EI sample, suggesting a distortion occurred during EI method[42]. Moreover, we also find that the EI can induce a lattice expansion of BTO. The distortion parameters are calculated as 0.012 for CP-PBf and 0.042 for EI-PBf (see Fig. 4j). As reported[41], the higher distortion level in EI-PBf can induce a higher performance in the piezoelectric of BTO.

In summary, we illustrated the mechanism of the mechanical poling in Fig. 4g. Firstly, the chain alignment during melt-state energy implantation can induce the crystallization of β-phase PVDF, which can contribute a d33 of -10.20 pC/N, in this work. Besides, the implanted energy can induce the crystalline orientation and lattice expansion of BTO, contributing to higher piezoelectric properties. Ultimately, we could obtain an advanced flexible piezoelectric polymeric composite with d33 of -51.20 pC/N, without the requirement of electrical-poling.

### Applications of PBf fabricated via EI method

In the above sections, we confirmed that our EI-PBf can be a highly sensitive self-powered pressure sensor. In addition, an energy harvesting experiment was conducted. The device was assembled as illustrated in Fig. 5a. During the experiment, a liner motor equipped with a force sensor was employed for performing mechanical deformation (10 N force applied), and the data was collected for analysis. It should be mentioned that due to the different materials used in the liner motor (Al) and the surface cover of the device (PI), there are unavoidable triboelectric signals collected, thus, the energy harvesting performance of PBf performed in this section consists not only piezoelectric effect, but also triboelectric contribution. Figure 5b, c show the voltage and power density of both CP-PBf and EI-PBf, respectively, at different resistances. From the data, we can see that along with the higher resistance used, the $V_{oc}$ increases with higher resistance. The highest output power obtained in this work is ~42.57 mW/m$^2$ and ~58.72 mW/m$^2$. In addition, the durability is tested as shown in Fig. 5d–f. From the data, we can see that our sample shows a high constant transferred charge in a long period (12,000 s), which evidences the durability of our flexible and lightweight polymeric piezoelectric. Moreover, we also performed a proof of concept for the application of our EI-PBf as a human motion sensor in Fig. 5g. In this section, several normal motions including back strike, squat up and high five, were performed, as well as the output voltage monitored. From the data, we can see that the output voltage varies from ~40 V to ~90 V, depending on the motion types. Moreover, different movements can induce different patterns of voltage, validating the effectiveness of the PBf as a self-powered human motion sensor.

## Discussion

Piezoelectric materials are now widely used in practical applications. However, the state-of-the-art in fabricating high-performance piezoelectric material relies on the time-consuming and complex electrical-poling process, which impedes its application. In this work, we proposed a concept of energy implantation, which can pole the material during fabrication, through introducing melt-state dynamic pressure. The PBf fabricated via our method shows a high d33 of -51.20 pC/N, and g33 of -1610.5 mVm/N. Moreover, as compared with the other piezoelectric materials reported in previous literature, our sample shows the best FOM and $g_{33}^{ds}$, which evidences that our method can not only avoid the conventional used electrical-poling process but also is capable of fabricating a high-performance piezoelectric material.

The mechanism of energy implantation-induced self-poling was also revealed. From analysis, the implanted energy can promote the formation of β-crystal PVDF due to its higher packing density, which contributes to a d33 of -10.20 pC/N. Then, for the BTO phase, the energy induces its crystalline structure alignment with the (110) plane in the in-plane direction, and the lattice expansion observed in the BTO crystalline also induces its piezoelectric property enhancement. As compared with the contribution of PVDF, we found that the structure evolution of BTO is more pronounced.

In this work, we provide a concept from an energy viewpoint during polymer processing that can pole flexible piezoelectric polymers. We believe such a method can not only be used in fabricating high-performance self-poled piezoelectric materials but also expand its function in structure engineering of both polymers and ceramics.

## Methods

### Materials

PVDF used in this work was purchased from Shanghai 3F (916 F). BTO powders were obtained from Shanghai Aladdin (B106130) with purity of >99.5%. NaCl was a commercially available table salt that was purchased from the local market.

### Preparation of polyvinylidene difluoride/barium titanate foam (PBf)

In this paper, PBf samples were fabricated by a two-step method. In the first step, weighted PVDF, BTO, and NaCl (weight ratio 21:9:70 wt%) were melt blended using a Brabender mixer at 195 °C to achieve a fine distribution and dispersion of BTO and NaCl in the PVDF matrix. The PVDF/BTO/NaCl composite was compressed into thin sheets with 1 mm thickness at 195 °C. In the process of machining, a custom-made compression molding machine was utilized with its programmable logic controlling system was modified to control the moving plate down and up for designed compressing and releasing (the force-time relationship was shown in Fig. 4a). For every process, 120 repeats of the unit were conducted to perform the samples based on our pre-examination. For reference, constant pressure was used to compress the 1-mm thickness sample.

### Preparation of constant pressure-PBf and energy implantation-PBf (CP-PBf and EI-PBf)

The PBf was fabricated by immersing all composites (4*4*0.1 cm$^3$ sheet) in DI water at 70 °C for 5–7 days, in order to get rid of the salts and form the porous structure. The mass of all composites is about the same as that of 30 wt% before immersion, which proves that all NaCl has been removed. By changing the compression and release time, the samples were named as CP for constant pressure and EI for our samples.

### Characterizations

A scanning electron microscope (SEM, S-3700, Hitachi) was used to analyze the morphology of fresh fracture surfaces of all composites. Fourier-transform infrared (Nicolet 6700 IR spectrometer, Thermo Scientific Company) was used to characterize the crystalline structure of all composites. The composite material is measured in attenuated total reflection (ATR) mode with a sweep range of 4000–600 cm$^{-1}$. An X-ray diffractometer (PANalytical B.V., Xpert Powder) was used to analyze the structure of composites. The scanning angle and scanning speed of XRD were 5–60° and 2.4°/min, respectively. The structural refinement software Fullprof was used to analyze the structural changes and atomic shifts of BTO in composites. The porosity was measured by the density method, where the densities were all tested by the Solid-liquid Densimeter (ET-320D, Etnaln Company). A dielectric impedance spectroscopy was used to characterize the dielectric properties of the composites from $10^2$ to $10^8$ Hz at room temperature. A differential scanning thermal analyzer (Netzsch DSC 204 F1 Phoenix) is used to measure the transition temperature of composite samples. In the piezoelectric response measurement, an electrometer (6517B, Keithley) was employed to monitor the transferred charge, open-circuit voltage, and close-circuit current between the two electrodes on samples under a repeat compress that was induced by a linear motor

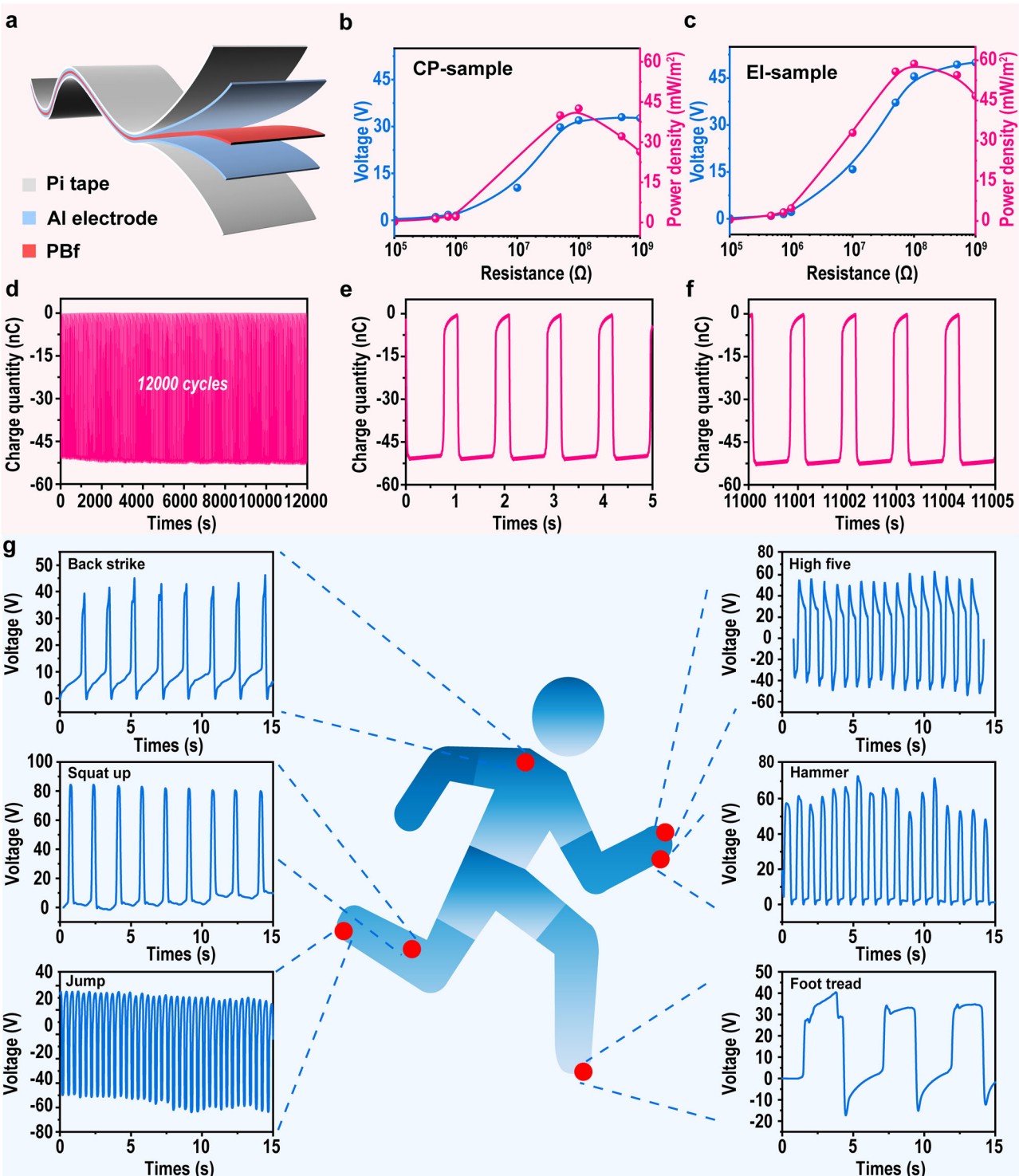

**Fig. 5 | Applications of PBfs. a** Illustration of the PBf-based device, containing PBf as a middle layer that is covered with Al electrodes on both sides and the surfaces were covered with PI tape. **b, c** The voltage and power density of the device pressed under 10 N with different resistances. **d, e, f** Long-term durability tests of our EI-PBf device under 10 N press. **g** Illustration of the voltage variations of the EI-PBf device under various human motions.

(LinMot). During this process, the force applied to the samples was measured and recorded via a digital force gauge (ZTS-i-HPO 15, IMADA).

## Data availability

The data supporting the findings of this study are reported in the main text or the Supplementary Information. Raw data are provided as a Source Data file. Source data are provided in this paper. All other data are available from the corresponding author upon request. Source data are provided in this paper.

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

## Acknowledgements

This work is supported by the National Natural Science Foundation of China (Grant No. 52103027, 51933007), and the Guangdong Basic and Applied Basic Research Foundation (Grant No. 2020A1515110467). Z.-X.H. also acknowledges the financial support from the opening project of the Guangdong Provincial Key Laboratory of Technique and Equipment for Macromolecular Advanced Manufacturing, South China University of Technology, China.

## Author contributions

Conceptualization: Z.-X.H., J.-P.Q. Methodology: Z.-X.H., L.-W.L., Y.-Z.H., W.-X.R. Investigation: Z.-X.H., L.-W.L., Y.-Z.H., W.-X.R., H.-W.J., J.W. Visualization: Z.-X.H., L.-W.L. Validation: Z.-X.H., L.-W.L., H.-W.J., J.W., H.-H.Z. Funding acquisition: Z.-X.H., H.-Z. H., J.-P.Q. Project administration: Z.-X.H., J.-P.Q. Supervision: Z.-X.H., H.-Z.H., J.-P.Q. Writing—original draft: Z.-X.H., L.-W.L. Writing—review and editing: All authors

## Competing interests

The authors declare no competing interests.
