## [Peer Review File · Nature Communications]

Self-poled Piezoelectric Polymer Composites via Melt-state Energy ImplantationReviewers' Comments:

Reviewer #1:

Remarks to the Author:

In this work, the authors developed a melt-state dynamic pressure procedure for fabricating the self-poled piezoelectric composite. The authors mixed PVDF with BTO to form a polymer composite which shows a d_{33} of ~ 51.20 pC/N at a density of 0.64 g/cm³. To show its practical impact, the PVDF/BTO composite can also serve as a pressure sensor and mechanical energy harvester. As far as I am concerned, this is a routine story since there are more than 1000+ piezoelectric composite reports in the community using PVDF/BTO.

Moreover, in the authors' previous report (Nano Energy 2022, 104,107921), they already showed that through selectively localizing BTO on the pore surface of open-cellular structured PVDF (PVDF/BaTiO₃ foams (PBfs), exactly the same materials name as the current report), they could largely enhance its piezoelectric output with a low density.

In summary, with a careful reading of this paper, the idea presented is not new and the practical demonstrations are routine. The reviewer does not see much scientific advancement in the current report that deserves a publication in Nature Communications.

Reviewer #2:

Remarks to the Author:

The article entitled "Self-poled piezoelectric polymer composites via melt-state energy implantation" describes the formation of a BTO-PVDF porous composite via a novel cyclic hot press moulding approach. The manuscript is well presented, however contains major issues that should be addressed prior to publication; most notably:

1) The system is a hybrid energy harvesting device with $> 60\%$ porosity (as described in Figure S3); this means the contribution from ferroelectrets must be highlighted and discussed in the manuscript;

2) The follow on for this is when benchmarking the sample against literature – this needs to also be done against porous PVDF structures; not just self-poled or otherwise PVDF solid films;

3) The measurement of charge vs force needs to be explained in more detail with applied force/displacement waveforms shown to provide strong support for the piezoelectric interpretations within the manuscript.

Beyond this, the mechanistic studies are well done (Figure 4) and the application (Figure 5) is a nice addition.

Please find further comments below:

- Can the authors please provide the time domain of the dynamic pressure cycling shown in Figure 1c? How does the strain rate provided by the authors technique compare to hot drawing PVDF to produce a high beta-phase fraction (<https://doi.org/10.1002/pat.5268>)?

- The beta phase fraction of 26% or EI-PBf (Figure 2c) seems quite high for the Raman data shown in Figure 2b and the XRD data shown in Figure 2e. The method of beta phase calculation in the supplementary text is for Raman – yet the only fitted beta peak shown is from XRD. Can the authors please describe how this value was calculated?

- What was the total crystallinity (of the polymer) of the CP-PBf vs the EI-PBf samples

- How many samples were tested for each loading/unloading cycles? Can the authors please provide

error bars on Figure 2c; Figure 3b; Figure 4b; Figure 4c; Figure 4f.

- The linear fit for EI-PBf in Figure 3b does not appear appropriate; it looks like there are two distinct regimes; between 10 and 30N where pores are being compressed giving rise to induction from ferroelectrets (<https://doi.org/10.1016/j.nanoen.2018.12.040>; <https://doi.org/10.1016/j.nanoen.2018.05.016>); and >30N where the polymer/BTO composite is being compressed giving rise to piezoelectric only contributions. The fit for this plot should be revised.

- The measured d_{33} is much lower than prior reports of ferroelectret based PVDF energy harvesters (<https://pubs.rsc.org/en/content/articlehtml/2019/sm/c8sm02160k>); why is the g_{33} shown in Fig 3 e show much higher here? Have the authors referenced ferroelectret + piezoelectric PVDFs or are the g_{33} values in Figure 3e all solid films?

- Please add the references used for Figure 3 e & f into the caption;

- The measurement of 'piezoelectric' properties using a hammer as described in the supplementary information is not correct; even adopting the polarity inversion method of reference 17 in the literature. Can the authors please provide the force, separation, and displacement used during these measurements. Leon et al. recently showed how even small separations lead to triboelectric contributions during measurements (<https://doi.org/10.1016/j.nanoen.2023.108445>); as did Sutka et al. (<https://doi.org/10.1002/adma.202002979>) – while FFT may not present a single harmonic due to the presence of pores and ferroelectret components in these samples; separation between the mechanical testing device and the sample being examined for piezoelectric properties should be avoided.

Reviewer #3:

Remarks to the Author:

The manuscript submitted by Huang et al. discusses "Self-poled Piezoelectric Polymer Composites via Melt-state Energy Implantation". In this manuscript, the authors present a novel concept of energy implantation on the β -phase of PVDF as well as on the orientation of the BTO crystal, which improves the piezoelectric properties of PVDF/BTO. The work is interesting but it requires some modifications.

1. Pg.4, Ln 70, The authors prepared EI-PBf by applying pressure cyclically at 195°C. Did the pressure continue to be applied during the cooling process? If not, will lowering the pressure during the cooling process decrease the formation of the PVDF crystalline phase?
2. Pg. 9, Ln 163, the authors claimed that the d_{33} value of 51 pC/N when T_{off} is 1.2 s, which is ~50% higher than a normal d_{33} values. Is this arise from the effect of T_{off} on the β -phase in PVDF? Later in Pg. 14, the authors stated in Fig. 4(f) that BTO is the main contributor to the d_{33} value of PVDF/BTO sample. Why does the different T_{off} have such a big effect on the d_{33} value? The authors need to give more explanation on this.
3. Pg. 11, the densities and porosities of all PBfs manufactured at different T_{off} are discussed in Figures S2 and S3, but the effect of pore sizes also should be discussed for these parameters.
4. Pg. 10, The authors tested the V_{oc} values at different pressures in Fig. 3(c) and noted that the V_{oc} value reaches ~20 V with a pressure of 86.6 kPa. Is it possible to increase the V_{oc} value if the pressure value continues to increase, and what is the maximum external pressure that can be achieved?
5. The authors mention "PVDF/BiTiO₃" in Table S2, please reconfirm whether it is "BaTiO₃" or "BiTiO₃".
6. Caption of Figure 2 g and 2 h should match the previous format for other figures.
7. A format error of " 6.1×10^4 and 7.7×10^4 cm² mol⁻¹" has occurred in the PVDF (β) calculation in the supplementary information.
8. The following papers should be cited:

S Zhang, et al. ACS Applied Materials & Interfaces, 13(27), 32242, 2021.
C. Wang, et al, Advanced Science, 8, 8, 2021.

We would like to thank the reviewers for their valuable comments and suggestions on this manuscript, which greatly improves the quality of our manuscript. Following these comments and suggestions, we have made careful revisions to our previous manuscript, and provide response to the comments point-by-point as follows:

Reviewer #1:

COMMENT 1#: In this work, the authors developed a melt-state dynamic pressure procedure for fabricating the self-poled piezoelectric composite. The authors mixed PVDF with BTO to form a polymer composite which shows a d_{33} of ~ 51.20 pC/N at a density of 0.64 g/cm³. To show its practical impact, the PVDF/BTO composite can also serve as a pressure sensor and mechanical energy harvester. As far as I am concerned, this is a routine story since there are more than 1000+ piezoelectric composite report in the community using PVDF/BTO.

Moreover, in the authors' previous report (Nano Energy 2022, 104,107921), they already showed that through selectively localizing BTO on the pore surface of open-cellular structured PVDF (PVDF/BaTiO₃ foams (PBfs), exactly the same materials name as the current report), they could largely enhance its piezoelectric output with a low density.

In summary, with a careful reading of this paper, the idea presented is not new and the practical demonstrations are routine. The reviewer does not see much scientific advancement in the current report that deserves a publication in Nature Communications.

Response:

We would like to appreciate the valuable comment from reviewer. We agree that PVDF/BTO is widely employed as piezoelectric material and well investigated material. In **Figure R1**, we showed the publication number of PVDF/BTO piezoelectrics that was collected from Web of Knowledge with keywords “PVDF” and “BaTiO₃”. From data, we can see there is a clear trend of increasing interest in fabricating PVDF/BTO piezoelectrics with outstanding properties.

Figure R1. Publications of each year in area of PVDF/BTO. (data obtained from Web of Knowledge with keywords “PVDF” and “BaTiO₃”)

Different with these reports, the main purpose of this work is to demonstrate a novel method to avoid the time and energy-consuming electrical-poling process for piezoelectric materials. Thus, we think using a widely investigated material to perform this work is more proper.

In addition to the simple fabrication method and avoiding of electrical-poling process, we also prepared a comparison of d_{33} values between our sample and recent reports¹⁻¹² in **Figure R2** to show the advantages of energy implantation.

Figure R2. Comparison of d_{33} value between our work and several recent reports.

As been mentioned by the reviewer, we have done several work in fabricating high performance piezoelectric polymer composites. Previously, as been cited in the original manuscript (ref 12), we found that through engineering the pore structure of PVDF/BTO foams and the selectively localization of BTO, the piezoelectric output of electrical-poled samples can be largely enhanced. And we also demonstrated that the reason for such enhancement can be attributed to the space charge electret phenomenon during electrical poling. However, it should be pointed out that the piezoelectric output still relies on time-consuming electrical poling.

In addition to ours' report, several recent work also show that although the materials and structure of piezoelectric materials can be optimized for better piezoelectric performance, the electrical poling is still necessary. For instance, Kim et al.¹³ reported a 14-hour electrical poling is required for PVDF/BaTiO₃ to achieve a high piezoelectric output; Guan et al.¹⁴ showed that the P(VDF-TrFE)/BaTiO₃ composites can perform a 6 V and 1.5 μ A piezoelectric output after 12-hour poling under 50 MV/m electrical field.

From the sustainable development point of view, we then moved our focus to self-poled piezoelectric materials that can present piezoelectric output without electrical poling. In this work, we purposed a novel concept of energy implantation during melt-state fabrication that can pole PVDF/BTO composites. The none-electrical poled composites can show a high d_{33} of ~51.20 pC/N, which even higher than many reported values of electrical-poled composites. Moreover, we also provided the physical insight besides the novel energy implantation induced poling phenomenon.

Meanwhile, we also prepared a comparison between the PBfs fabricated *via* our energy implantation

concept (this work) and conventional method (our previous work) in **Figure R3**. From the comparison, it is clear that in addition to the advantage of non-electrical poling, the piezoelectric performance of this work is much better than our previous work.

Figure R3. Comprehensive comparison of the properties between this work and our previous work.

Thus, we believe that although a novel material was not used in this work, but the novel method we proposed in this work that suggesting energy implantation can induce self-poling of polymer composites to show outstanding piezoelectric output could provide novel insight in polymer engineering society for advanced piezoelectrics fabrication.

To enhance the novelty of this work, we have modified the introduction section in the revision.

Reviewer #2:

The article entitled “Self-poled piezoelectric polymer composites via melt-state energy implantation” describes the formation of a BTO-PVDF porous composite via a novel cyclic hot press moulding approach. The manuscript is well presented, however contains major issues that should be addressed prior to publication; most notably:

Response:

The authors acknowledge the constructive comment from reviewer, which would encourage us continue our passion in this area.

COMMENT 1#: The system is a hybrid energy harvesting device with > 60% porosity (as described in Figure S3); this means the contribution from ferroelectrets must be highlighted and discussed in the manuscript;

Response:

We appreciate the valuable comment from reviewer. As been commented, ferroelectrets is one of major sort in porous piezoelectric materials, that have been widely investigated and practical applied. As a porous material, the voids can be charged by subjecting ferroelectret to a high electric field that can generate microplasma discharges inside¹⁵. Consequently, the opposite internal surfaces of void would carry positive and negative electric charges and form the ferroelectret. And during deformation, the ferroelectrets can serve as dipoles to present superior piezoelectric performances (usually, the d_{33} ranges in 100-500 pC/N).

From the mechanism of ferroelectret (see **Figure R4**), it can be seen that the high electric field poling is essential for charging the voids. In our previous work (ref 12), we also studied the contribution from ferroelectret in the electrical-poled porous PVDF/BTO composites. The results showed that through pore structure engineering, the contribution from ferroelectret can be largely enhanced due to the electrical-poling process can induce more charges stored on the surfaces of pores.

In this work, it should be mentioned that the main purpose is to report a novel method that can fabricate high performance piezoelectric material without electrical-poling process. Thus, although our PVDF/BTO composites have a high fraction of pores, they cannot serve as ferroelectrets due to no electric field applied to the formation of charges.

In revision, related information has been included for enhancing the quality of our manuscript.

Figure R4. Schematic of the working mechanism of ferroelectret. (a) Townsend breakdown under a high poling voltage. (b) charged polymer samples having positive/negative electrical charges with the polarization of \vec{P}_s . (c) piezoelectrically induced charge flow when external force applied along the polarisation direction.

COMMENT 2#: The follow on for this is when benchmarking the sample against literature – this needs to also be done against porous PVDF structures; not just self-poled or otherwise PVDF solid films;

Response:

The authors acknowledge the constructive comment from reviewer. In revision, we have included several recently reported porous polymeric composites for comparison, in **Figure S16** and **Table S2**.

Moreover, we also revised the manuscript with more discussion of porous polymer/ceramic composites in the revision.

Figure S16. Comparison of the g_{33} value between our self-poled sample and reported electrical poled polymer/ceramic hybrid ferroelectrets

COMMENT 3#: The measurement of charge vs force needs to be explained in more detail with applied force/displacement waveforms shown to provide strong support for the piezoelectric interpretations within the manuscript;

Response:

Thanks for your valuable comment. Herein, to clearly show the measurement, the photographs were taken for our setup as shown in **Figure S9a**. Moreover, the displacement and force vs time curves were also recorded and plotted in **Figure S9b**.

Related information has been added in the revision.

Figure S9. (a) Photographs of the setup for piezoelectric tests, and (b) Applied force / displacement waveform for the piezoelectric tests.

Beyond this, the mechanistic studies are well done (Figure 4) and the application (Figure 5) is a nice addition.

Please find further comments below:

COMMENT 4#: Can the authors please provide the time domain of the dynamic pressure cycling

shown in Figure 1c? How does the strain rate provided by the authors technique compare to hot drawing PVDF to produce a high beta-phase fraction (<https://doi.org/10.1002/pat.5268>)?

Response:

We totally agree that the time domain of the dynamic pressure cycling is essential for readers to understand and repeat our work. In the manuscript, **Figure 1c** is just an illustration of the dynamic pressure process for readers to understand the main thought of energy implantation, it does not represent the real pressure-time relationship. As shown in **Figure 1c**, each dynamic pressure cycle contains two stages: pressure-on and pressure-off, to realize a higher energy implantation. In the pressure-on stage, the pressure applied on the material raises from 0 MPa to a set value (P_{peak}) and then immediately drops to 0 MPa. Then, the process goes into pressure-off stage, in which no pressure is applied for a certain time (T_{off}). In this work, the main variation is the T_{off} .

In revision, we have modified the caption of **Figure 1c** to avoid any misunderstanding.

The pressure-time relationships of different samples were shown in **Figure 4a**, in which the T_{off} ranges from 0 s to 2.10 s. Above information is also shown in **Figure R5**.

Figure R5. (a) Force-time relationship of one cycle in our melt-state dynamic pressure process. (b) The pressure-time relationship of repeat unit in EI-method with different T_{off} .

We thank the review for commenting in the strain rate effect. Before comparing the strain rate between ours and the reported value, it should be mentioned that mechanical deformation that our sample experienced was compression, while the reference sample was stretching. The reference work¹⁶ reported ~50% content of β -phase PVDF can be obtained *via* stretching with strain rate of 0.02 s^{-1} . As been widely accepted that the strain rate cannot be set as a high value, for the purpose of achieving high stretching rate without failure. While, for our method, compression was introduced with a strain rate of 1.16 s^{-1} , which is much higher than the stretching. We believe that the higher strain rate may also has contribution for the β -phase PVDF formation.

We would also acknowledge the reviewer for pointing us a new and interesting topic that what is the difference between compression and stretch for the crystalline structure control of PVDF. We would like to take some experiments and prepare our report in future.

Related information was updated in the revision.

COMMENT 5#: The beta phase fraction of 26% or EI-PBf (Figure 2c) seems quite high for the Raman data shown in Figure 2b and the XRD data shown in Figure 2e. The method of beta phase calculation in the supplementary text is for Raman – yet the only fitted beta peak shown is from XRD. Can the authors please describe how this value was calculated?

Response:

Thanks the reviewer for pointing out this important comment. Generally, the crystal structure of PVDF can be distinguished and calculated from FTIR, DSC and XRD. As can be found in the review by Martins et al.¹⁷, the β -phase PVDF can be calculated as

$$F(\beta) = \frac{A_{\beta}}{\left(\frac{K_{\beta}}{K_{\alpha}}\right)A_{\alpha} + A_{\beta}} \times 100\%$$

Where A_{α} and A_{β} are the absorbance at 764 and 840 cm^{-1} , respectively. K_{α} and K_{β} are the corresponding absorbance coefficients, 6.1×10^4 and $7.7 \times 10^4 \text{ cm}^2 \text{ mol}^{-1}$, respectively.

And the fraction of γ phase can be calculated as:

$$F(\gamma) = \frac{A_{\gamma}}{\left(\frac{K_{\gamma}}{K_{\alpha}}\right)A_{\alpha} + A_{\gamma}} \times 100\%$$

Where A_{α} and A_{γ} are the absorbance at 764 and 832 cm^{-1} , respectively. K_{α} and K_{γ} are the corresponding absorbance coefficients, 0.365 and 0.150 μm^{-1} , respectively. $F(\alpha)$ is calculated using $1 - F(\beta) - F(\gamma)$.

From these equations, we can see that the characteristic peak of β -phase and γ -phase are 840 cm^{-1} and 832 cm^{-1} , which leads to some confusions in distinguishing them. Thus, XRD is usually employed to provide further information of the crystal structures. Thus, in this work, we used both FTIR and XRD to quantitatively analyzing the fraction of each crystal structure of PVDF.

Related information has been added in the revision.

COMMENT 6#: What was the total crystallinity (of the polymer) of the CP-PBf vs the EI-PBf samples ?

Response:

We totally agree that the crystallinity of polymer plays a central role in determining its properties. Thus, the crystallinities of both CP-PBf and EI-PBf were measured and calculated in the revision. Besides, considering there are several methods can be used for characterizing the crystallinity of polymers, two methods were employed and compared herein.

Figure S8 shows the crystallinities of all samples fabricated in this work, by DSC and XRD. It

should be mentioned that, as been widely investigated, there are difference in the crystallinities calculated by DSC and XRD. But the trends in the crystallinities by DSC and XRD are same. With a higher Toff used in the dynamic pressure, the crystallinity decreases from 0 s Toff to 1.20 s Toff and then increases, suggesting that the dynamic pressure can influence the crystallization behavior of PVDF.

Below shows the methods for calculation of crystallinities:

(1) DSC

The degree of crystallinity (χ_c) of CP-PBf and EI-PBf samples can be calculated by the following equation:

$$\chi_c = \frac{\Delta H_m}{\Delta H_0} \times 100\%$$

where ΔH_m and ΔH_0 represent melting enthalpies of the present sample and of perfectly crystalline PVDF (104.5 Jg^{-1})^{18, 19}, respectively.

(2) XRD

The variations of crystallinity that was calculated from the following equation based on XRD patterns.

$$\chi_c = \frac{S_c}{S_c + S_a} \times 100\%$$

where, S_c and S_a represent the sum of areas of crystalline parts and amorphous parts in XRD diffraction peaks, respectively. Peak separation and area calculation were all done through JADE²⁰.

Figure S8. Crystallinity of CP-PBf and EI-PBf Samples calculated by (a) DSC and (b) XRD

COMMENT 7#: How many samples were tested for each loading/unloading cycles? Can the authors please provide error bars on Figure 2c; Figure 3b; Figure 4b; Figure 4c; Figure 4f.

Response:

Thank you for your constructive comment. For all samples in this work, 3 repeats were taken for

the repeatability. In revision, we added error bars on each figure. Especially considering the most essential achievement in this work is the high d33 value obtained, we also included the charge-force relationships of each repeat in the revision, as shown in **Figure S10**.

Figure S10. Charge-force relationships of each repeat.

COMMENT 8#: The linear fit for EI-PBf in Figure 3b does not appear appropriate; it looks like there are two distinct regimes; between 10 and 30N where pores are being compressed giving rise to induction from ferroelectrets (<https://doi.org/10.1016/j.nanoen.2018.12.040>; <https://doi.org/10.1016/j.nanoen.2018.05.016>); and >30N where the polymer/BTO composite is being compressed giving rise to piezoelectric only contributions. The fit for this plot should be revised..

Response:

The authors acknowledge the reviewer for the valuable comment. The curve shown in **Figure 3b** of EI-PBf is one of three repeats. Below we showed all three repeats during the test in **Figure R6**. From the curves, we can see that all curves have good linear fits. Thus, we believe the two distinct regimes in the curve can be attributed to the measurement error.

For the ferroelectret contribution, as we discussed above, considering the abandon of electrical-poling process, our samples do not contain any ferroelectrets.

Related information has been included in the revision.

Figure R6. Three repeated test results and R-Square value of EI with $T_{\text{off}} = 1.20$ samples.

COMMENT 9#: - The measured d_{33} is much lower than prior reports of ferroelectret based PVDF energy harvesters (<https://pubs.rsc.org/en/content/articlehtml/2019/sm/c8sm02160k>); why is the g_{33} shown in Fig 3e show much higher here? Have the authors referenced ferroelectret + piezoelectric PVDFs or are the g_{33} values in Figure 3e all solid films?

Response:

Thanks for your comment for enhancing the quality of our work. We agree that the d_{33} of samples we fabricated in this work is much lower than prior reports of ferroelectret based PVDF, whose d_{33} usually ranges in 200-500 pC/N. This is due to that in ferroelectret, the voids would be charged with opposite charges in the different sides of the pore's surfaces, and then these voids play as large and easy-to-deform dipoles, and presenting high piezoelectric performance. However, it should be mentioned that all these ferroelectret require electrical-poling to perform piezoelectric behaviors. In this work, the main purpose is to develop a novel method to avoid the time and energy-consuming electrical-poling.

Besides, the g_{33} value shown in Figure 3e is density-specific g_{33}^{ds} , and can be calculated as:

$$g_{33}^{ds} = \frac{g_{33}}{\rho}$$

Where ρ is the density. Herein, we also provided all the piezoelectric performance data in the **Table S1**.

From the comparison, due to the lower density of our EI-PBf, its g_{33}^{ds} is higher than other reports. And as been mentioned by the reviewer, in the original manuscript, the data mostly came from solid samples. Herein, as suggested by the reviewer, we modified the comparison with several recently reported advanced porous polymer/ceramic hybrid piezoelectrics, as shown in **Figure S16**. Below, we also collected the data in **Table S2**. From the comparison, it is clear that although we could obtained a high d_{33} and g_{33} value, but it is still lower than the mentioned reference²¹, which is reasonable that the ferroelectrets have outstanding piezoelectric performance. However, comparing with this ferroelectrets, our samples can provide piezoelectric output without the time and energy-consuming electrical-poling process, which is beneficial for the sustainable development of this area.

Figure S16. Comparison of the g_{33} value between our self-poled sample and reported electrical poled ferroelectrets.

Table S2 Piezoelectric properties of our EI-PBf and other reported ferroelectrets materials.

	d_{33} (pc/N)	g_{33} (10^{-3}VmN^{-1})	FOM ($10^{-15} \text{m}^2\text{N}^{-1}$)	References
PS/KNNS-BNZFe	400	58	20887	22
PZT/PDMS	497	110	54218	23
CB/PVDF-HFP	58	260	15080	24
This work	51	1611	82541	

COMMENT 9#: Please add the references used for Figure 3 e & f into the caption

Response:

We sincerely acknowledge that reviewer for the careful review. In revision, we have added the references for comparison into the caption.

COMMENT 10#: The measurement of 'piezoelectric' properties using a hammer as described in the supplementary information is not correct; even adopting the polarity inversion method of reference 17 in the literature. Can the authors please provide the force, separation, and displacement used during these measurements. Leon et al. recently showed how even small separations lead to triboelectric contributions during measurements (<https://doi.org/10.1016/j.nanoen.2023.108445>); as did Sutka et al. (<https://doi.org/10.1002/adma.202002979>) – while FFT may not present a single harmonic due to the presence of pores and ferroelectret components in these samples; separation between the mechanical testing device and the sample being examined for piezoelectric properties should be avoided.

Response:

Thanks for your constructive comment. As been mentioned by the reviewer, how to avoid triboelectric contribution during piezoelectric test is essential in evaluating a piezoelectric. And we totally agree that revisiting the data of our tests and validating with other methods is very important in enhancing the quality of our work.

As demonstrated by Wang et al.²⁵, the method used herein for piezoelectric performance measurement did not try to avoid the triboelectric. While, on the contrary, through subtracting the output charges at same applied pressure, the contribution of triboelectric can be eliminated. In this work, the effectiveness of this method was validating by a commercial piezoelectric PVDF film and a non-piezoelectric PET film. More details can be found in **Supporting Information** and Wang’s work.

In addition, considering that the measurement of d_{33} is essential for our work, to further confirm that we obtained a believable d_{33} value, another method with no separation during test was employed (as been suggested by the reviewer). **Figure S13** shows the charge vs force results, and suggesting a d_{33} value of 52.7 pC/N, similar with the value we reported (51.2 pC/N). In this test, the force applied on the specimen was controlled from ~2N to a certain value, and the variations in force was used for d_{33} calculating.

Figure S13. (a) The charge-force relationship of our EI-PBf measured using a separation method and a non-separation method, and (b) applied force / displacement waveform for the non-separation piezoelectric tests.

Related information was added in the revision.

Responses Reviewer #3:

The manuscript submitted by Huang et al. discusses “Self-poled Piezoelectric Polymer Composites via Melt-state Energy Implantation”. In this manuscript, the authors present a novel concept of energy implantation on the β -phase of PVDF as well as on the orientation of the BTO crystal, which improves the piezoelectric properties of PVDF/BTO. The work is interesting but it requires some modifications.

Response:

The authors acknowledge the constructive comment from reviewer, which would encourage us continue our passion in this area.

COMMENT 1#: Pg. 4, Ln 70, The authors prepared EI-PBf by applying pressure cyclically at 195°C. Did the pressure continue to be applied during the cooling process? If not, will lowering the pressure during the cooling process decrease the formation of the PVDF crystalline phase?

Response:

Thank you for the valuable comment. As mentioned by reviewer, pressure (25 MPa) was applied during the cooling process. But, unlike the melt-state with dynamic pressure, the pressure applied during cooling process is a constant value of 25 MPa. Same condition is used in the CPM samples. The employment of pressure during cooling is to avoid the structure relaxation of PVDF molecular chains during cooling and maintain the history of melt-state thermos-mechanical process.

As for another comment from reviewer, the effect of dynamic pressure on the formation of b-phase PVDF has been systemically investigated in our previously work²⁶. In that work, the pressure, dynamic condition, temperature and thickness of sample were all investigated. Thus, in this work, to enhance the logic, our main focus is on the structure variations of both PVDF and BTO and their effects on the piezoelectric properties.

Related information has been added in revision.

COMMENT 2#: Pg. 9, Ln 163, the authors claimed that the d_{33} value of 51 pC/N when T_{off} is 1.2 s, which is ~50% higher than a normal d_{33} values. Is this arise from the effect of T_{off} on the β -phase in PVDF? Later in Pg. 14, the authors stated in Fig. 4(f) that BTO is the main contributor to the d_{33} value of PVDF/BTO sample. Why does the different T_{off} have such a big effect on the d_{33} value? The authors need to give more explanation on this.

Response:

We acknowledge the reviewer for focusing on one of the most important achievements in this work. Investigating the mechanism for energy implantation induced piezoelectric enhancement is essential, and in this work, we showed the mechanism in **Figure 4**. To quantitatively analyzing the

contribution from PVDF and BTO for the d33 enhancement, we measured EI-samples with and without BTO. As can be seen in **Figure 4f**, the d33 of EI-PVDF is 10.2 pC/N, due to the energy implantation induced b-phase formation of PVDF. In addition, the d33 of EI-PVDF/BTO sample is 51.20 pC/N. Therefore, the further improvement in d33 is attributed to the structure evolution of BTO.

Moreover, we also analyzed the reason for the PVDF and BTO's contributions for d33. For PVDF, its contribution for d33 enhancement is believed due to the formation of b-phase crystal, which has been widely investigated. For BTO, as can be seen in **Figure R7**, we performed the atomic position of BTO. From data, it is clear that the EI process can significantly affect the structure of BTO. When the T_{off} was set as 1.20 s, the positions of Ti and O2 shifted obviously, which leads to a higher level of structure disorder, and can present higher piezoelectric output. While, for the reason why 1.20 s T_{off} could induce such a large variation of BTO, we think it is due to the energy implanted into material at this T_{off} surpass a threshold. Further work is still required in future to gain deeper insight.

Related information has been included in the revision.

Figure R7. The atomic positions of BTO corresponding to different processes.

COMMENT 3#: Pg. 11, the densities and porosities of all PBfs manufactured at different T_{off} are discussed in Figures S2 and S3, but the effect of pore sizes also should be discussed for these parameters.

Response:

Thanks for your valuable comment. In revision, we performed pore size analysis via SEM images. For each sample, 400 counts were chosen and measured that were randomly picked from at least 4 SEM images. The results were plotted in **Figure S4**, which suggested that the dynamic pressure has little influence in the pore size of as-prepared samples.

Related information has been added in the revision.

Figure S4. The pore size analysis of all samples fabricated in this work.

COMMENT 4#: Pg. 10, The authors tested the Voc values at different pressures in Fig. 3(c) and noted that the Voc value reaches ~20 V with a pressure of 86.6 kPa. Is it possible to increase the Voc value if the pressure value continues to increase, and what is the maximum external pressure that can be achieved?

Response:

Thanks for your valuable comment. We agree that exploring the maximum external pressure that can be achieved and the consequent electricity output is an interesting topic for our composites. Thus, in revision, the pressure applied was extended to 158.7 kPa. And from the data, we can see the relationship between Voc and applied pressure is still linear, suggesting that our sample can maintain a good piezoelectric output in a wide range of applied force. It should be mentioned that due to the limitation of the linear motor output is 160 N, thus, the maximum pressure was limited. And this may not reach the limit of our material. In future work, we will try to further employ higher pressure to check the consequent output.

In revision, related information has been modified.

Figure S14. The Voc of EI-PBf under higher applied pressures.

COMMENT 5#: The authors mention "PVDF/BiTiO₃" in Table S2, please reconfirm whether it is "BaTiO₃" or "BiTiO₃".

COMMENT 6#: Caption of Figure 2 g and 2 h should match the previous format for other figures.

COMMENT 7#: A format error of " 6.1×10^4 and $7.7 \times 10^4 \text{ cm}^2 \text{ mol}^{-1}$ " has occurred in the PVDF (β) calculation in the supplementary information.

Response:

We sincerely apology for our typos and mistakes during preparation of the original manuscript. In revision, all authors carefully reviewed the entire manuscript.

COMMENT 8#: The following papers should be cited:

S Zhang, et al. *ACS Applied Materials & Interfaces*, 13(27), 32242, 2021.

C. Wang, et al, *Advanced Science*, 8, 8, 2021

Response:

We thank the reviewer for bringing these important articles to our attention. We have now added and discussed the these references in the revised manuscript.

Reference

1. Yang, C.; Song, S.; Chen, F.; Chen, N., Fabrication of PVDF/BaTiO₃/CNT Piezoelectric Energy Harvesters with Bionic Balsa Wood Structures through 3D Printing and Supercritical Carbon Dioxide Foaming. *ACS Applied Materials & Interfaces* **2021**, 13 (35), 41723–41734.
2. Jiang, H.; Song, L.; Huang, Z.-X.; Liu, M.; Zhao, Y.; Zhang, S.; Guo, J.; Li, Y.; Wang, Q.; Qu, J.-P., A novel concept of hierarchical porous structural design on enhancing output performance of piezoelectric nanogenerator. *Nano Energy* **2022**, 104, 107921.
3. Patil, R.; Ashwin, A.; Radhakrishnan, S., Novel polyaniline/PVDF/BaTiO₃ hybrid composites with high piezo-sensitivity. *Sensors and Actuators A: Physical* **2007**, 138 (2), 361–365.
4. Athira, B. S.; George, A.; Vaishna Priya, K.; Hareesh, U. S.; Gowd, E. B.; Surendran, K. P.; Chandran, A., High-Performance Flexible Piezoelectric Nanogenerator Based on Electrospun PVDF–BaTiO₃ Nanofibers for Self-Powered Vibration Sensing Applications. *ACS Applied Materials & Interfaces* **2022**, 14 (39), 44239–44250.
5. Kubin, M.; Makreski, P.; Zanoni, M.; Gasperini, L.; Selleri, G.; Fabiani, D.; Gualandi, C.; Bu arovaska, A., Effects of nano-sized BaTiO₃ on microstructural, thermal, mechanical and piezoelectric behavior of electrospun PVDF/BaTiO₃ nanocomposite mats. *Polymer Testing* **2023**, 126, 108158.
6. Cho, Y.; Jeong, J.; Choi, M.; Baek, G.; Park, S.; Choi, H.; Ahn, S.; Cha, S.; Kim, T.; Kang, D.-S.; Bae, J.; Park, J.-J., BaTiO₃@PVDF–TrFE nanocomposites

with efficient orientation prepared via phase separation nano-coating method for piezoelectric performance improvement and application to 3D-PENG. *Chemical Engineering Journal* **2022**, *427*, 131030.

7. Li, L.; Guo, H.; Sun, H.; Sui, H.; Yang, X.; Wang, F.; Liu, X., The construction of BaTiO₃-based core-shell composites for high-performance and flexible piezoelectric nanogenerators. *Sensors and Actuators A: Physical* **2023**, *363*, 114553.
8. Mirjalali, S.; Bagherzadeh, R.; Mahdavi Varposhti, A.; Asadnia, M.; Huang, S.; Chang, W.; Peng, S.; Wang, C.-H.; Wu, S., Enhanced Piezoelectricity of PVDF-TrFE Nanofibers by Intercalating with Electrospayed BaTiO₃. *ACS Applied Materials & Interfaces* **2023**, *15* (35), 41806-41816.
9. Taleb, S.; Badillo, M.; Flores-Ruiz, F. J.; Acuatla, M., From synthesis to application: High-quality flexible piezoelectric sensors fabricated from tetragonal BaTiO₃/P(VDF-TrFE) composites. *Sensors and Actuators A: Physical* **2023**, *361*, 114585.
10. Chai, B.; Shi, K.; Wang, Y.; Liu, Y.; Liu, F.; Jiang, P.; Sheng, G.; Wang, S.; Xu, P.; Xu, X.; Huang, X., Modulus-Modulated All-Organic Core-Shell Nanofiber with Remarkable Piezoelectricity for Energy Harvesting and Condition Monitoring. *Nano Letters* **2023**, *23* (5), 1810-1819.
11. Yaseen, H. M. A.; Park, S., Enhanced Power Generation by Piezoelectric P(VDF-TrFE)/rGO Nanocomposite Thin Film. *Nanomaterials* **2023**, *13* (5), 860.
12. Choi, W.; Choi, K.; Yang, G.; Kim, J. C.; Yu, C., Improving piezoelectric performance of lead-free polymer composites with high aspect ratio BaTiO₃ nanowires. *Polymer Testing* **2016**, *53*, 143-148.
13. Kim, T.; Joshi, B.; Lim, W.; Samuel, E.; Aldalbahi, A.; Periyasami, G.; Lee, H.-S.; An, S.; Yoon, S. S., Scalable, flexible BaTiO₃/PVDF piezocomposites prepared via supersonic spraying for use in energy harvesting and integrated energy storage devices. *Nano Energy* **2023**, *115*, 108682.
14. Guan, X.; Xu, B.; Gong, J., Hierarchically architected polydopamine modified BaTiO₃@P(VDF-TrFE) nanocomposite fiber mats for flexible piezoelectric nanogenerators and self-powered sensors. *Nano Energy* **2020**, *70*, 104516.
15. Bauer, S.; Gerhard-Multhaupt, R.; Sessler, G. M., Ferroelectrets: Soft Electroactive Foams for Transducers. *Physics Today* **2004**, *57* (2), 37-43.
16. Men, S.; Gao, Z.; Wen, R.; Tang, J.; Zhang, J. M., Effects of annealing time on physical and mechanical properties of PVDF microporous membranes by a melt extrusion-stretching process. *Polymers for Advanced Technologies* **2021**, *32* (6), 2397-2408.
17. Martins, P.; Lopes, A. C.; Lanceros-Mendez, S., Electroactive phases of poly(vinylidene fluoride): Determination, processing and applications. *Progress in Polymer Science* **2014**, *39* (4), 683-706.
18. Lanceros-Méndez, S.; Mano, J. F.; Costa, A. M.; Schmidt, V. H., FTIR AND DSC STUDIES OF MECHANICALLY DEFORMED β -PVDF FILMS. *Journal of Macromolecular Science, Part B* **2001**, *40* (3-4), 517-527.
19. Liu, Z.; Maréchal, P.; Jérôme, R., D.m.a. and d.s.c. investigations of the β transition of poly(vinylidene fluoride). *Polymer* **1997**, *38* (19), 4925-4929.
20. Zhou, Y.; Liu, W.; Tan, B.; Zhu, C.; Ni, Y.; Fang, L.; Lu, C.; Xu, Z.

Crystallinity and β Phase Fraction of PVDF in Biaxially Stretched PVDF/PMMA Films *Polymers* [Online], 2021.

21. Zhang, Y.; Bowen, C. R.; Deville, S., Ice-templated poly(vinylidene fluoride) ferroelectrets. *Soft Matter* **2019**, *15* (5), 825–832.
22. Xue, H.; Jiang, L.; Lu, G.; Wu, J., Multilevel Structure Engineered Lead-Free Piezoceramics Enabling Breakthrough in Energy Harvesting Performance for Bioelectronics. *Advanced Functional Materials* **2023**, *33* (11), 2212110.
23. Xu, Q.; Wang, Z.; Zhong, J.; Yan, M.; Zhao, S.; Gong, J.; Feng, K.; Zhang, J.; Zhou, K.; Xie, J.; Xie, H.; Zhang, D.; Zhang, Y.; Bowen, C., Construction of Flexible Piezoceramic Array with Ultrahigh Piezoelectricity via a Hierarchical Design Strategy. *Advanced Functional Materials* **2023**, *33* (41), 2304402.
24. Li, Y.; Tong, W.; Yang, J.; Wang, Z.; Wang, D.; An, Q.; Zhang, Y., Electrode-free piezoelectric nanogenerator based on carbon black/polyvinylidene fluoride-hexafluoropropylene composite achieved via interface polarization effect. *Chemical Engineering Journal* **2023**, *457*, 141356.
25. Chen, C.; Zhao, S.; Pan, C.; Zi, Y.; Wang, F.; Yang, C.; Wang, Z. L., A method for quantitatively separating the piezoelectric component from the as-received “Piezoelectric” signal. *Nature Communications* **2022**, *13* (1), 1391.
26. Huang, Z.-X.; Wang, M.-M.; Feng, Y.-H.; Qu, J.-P., β -Phase Formation of Polyvinylidene Fluoride via Hot Pressing under Cyclic Pulsating Pressure. *Macromolecules* **2020**, *53* (19), 8494–8501.

Reviewers' Comments:

Reviewer #2:

Remarks to the Author:

The authors have made significant revisions to the manuscript, which I now believe to be suitable for publication.

The results presented in response to Comment #3 (schematic of experimental set up for piezoelectric testing) still show a non-contact mode approach; however I believe this doesn't impact the overall manuscript given the comparative non-contact mode vs contact-mode data provided in Figure S13 - although an understanding of why the charge vs force curves in Figure S13 have different shapes for contact and non-contact mode would benefit an understanding of the end applications of the device.

Reviewer #3:

Remarks to the Author:

1. Instead of measuring the piezoelectric coefficient d_{33} using the value of Charge vs Applied pressure, the value of d_{33} should also be measured using a d_{33} meter (piezo d_{33} test system). 2. Except that, the authors have answered the most of the reviewer's questions.

We would like to thank the reviewers for their valuable comments and suggestions on this manuscript, which greatly improves the quality of our manuscript. Following these comments and suggestions, we have made careful revisions to our previous manuscript, and provide response to the comments point-by-point as follows:

Reviewer #2:

COMMENT 1#: The authors have made significant revisions to the manuscript, which I now believe to be suitable for publication. The results presented in response to Comment #3 (schematic of experimental set up for piezoelectric testing) still show a non-contact mode approach; however I believe this doesn't impact the overall manuscript given the comparative non-contact mode vs contact-mode data provided in Figure S13 - although an understanding of why the charge vs force curves in Figure S13 have different shapes for contact and non-contact mode would benefit an understanding of the end applications of the device.

Response:

The authors are very grateful to the reviewer for the recognition of our work, which will inspire us to further research in this field, especially the understanding of the different shapes of charge-force curves in contact mode and non-contact mode.

Reviewer #3:

COMMENT 1#: Instead of measuring the piezoelectric coefficient d_{33} using the value of Charge vs Applied pressure, the value of d_{33} should also be measured using a d_{33} meter (piezo d_{33} test system). 2. Except that, the authors have answered the most of the reviewer's questions.

Response:

We agree that the d_{33} meter is useful for further validation of the d_{33} values. Herein, the sample was measured using a d_{33} meter with value of 43.0 pC/N, as shown in supplementary Fig. 14.

Supplementary Fig. 14. D33 meter test results of EI-PBf sample.